# Difference in Body Composition Patterns between Age Groups in Italian Individuals with Overweight and Obesity: When BMI Becomes a Misleading Tool in Nutritional Settings

**DOI:** 10.3390/nu16152415

**Published:** 2024-07-25

**Authors:** Antonino De Lorenzo, Leila Itani, Marwan El Ghoch, Paola Gualtieri, Giulia Frank, Glauco Raffaelli, Massimo Pellegrini, Laura Di Renzo

**Affiliations:** 1Section of Clinical Nutrition and Nutrigenomic, Department of Biomedicine and Prevention, University of Tor Vergata, Via Montpellier 1, 00133 Rome, Italy; delorenzo@uniroma2.it (A.D.L.); laura.di.renzo@uniroma2.it (L.D.R.); 2Department of Nutrition and Dietetics, Faculty of Health Sciences, Beirut Arab University, Riad El Solh, Beirut 11072809, Lebanon; l.itani@bau.edu.lb; 3Center for the Study of Metabolism, Body Composition and Lifestyle, Department of Biomedical, Metabolic and Neural Sciences, University of Modena and Reggio Emilia, 41125 Modena, Italy; massimop@unimore.it; 4PhD School of Applied Medical-Surgical Sciences, University of Tor Vergata, Via Montpellier 1, 00133 Rome, Italy; giulia.frank@ymail.com (G.F.); glauco.raffaelli@yahoo.it (G.R.)

**Keywords:** body composition, BMI, body fat, central obesity, DXA, lifespan, overweight, sarcopenia

## Abstract

Little is known about the changes in body composition (BC) in people with overweight or obesity. The aim of this study was to assess the differences in BC patterns in this population based on gender and age. A total of 2844 Italian adults of mixed gender and a body mass index (BMI) of ≥25 kg/m^2^ underwent a BC assessment by means of dual-energy X-ray absorptiometry (DXA). The sample was categorized into three age groups: ‘young’ (20–39 years), ‘middle’ (40–59 years), and ‘older’ (60–80 years) adults, after being matched by body weight and BMI. Males showed higher total body fat percentage (BF%) and a lower total lean mass (LM), progressively from the young to the middle to the older age groups, while females showed similar values for these total compartments between the three age groups. However, in both genders, participants in the middle and older groups were more likely to have a higher trunk fat percentage by +1.23% to +4.21%, and lower appendicular lean mass (ALM) by −0.81 kg to −2.63 kg with respect to the young group, indicating expression of major central adiposity and sarcopenia. While our findings underscore the limitations of BMI to detect these differences between age groups, the identification of new tools suitable for this aim is greatly needed in this population. Moreover, further investigation that clarifies the impact of these differences in BC patterns between gender and age groups on health outcomes is also required.

## 1. Introduction

Changes in body composition (BC) during aging have been well documented, which is characterized by an increase in the total body fat (BF) and a decrease in lean mass (LM) [1,2,3,4], which may occur due to physiological fluctuations in body weight but with no significant changes in body mass index (BMI) [5]. This model has been validated in normal-weight individuals [6]. Obesity is a chronic disease defined as increased accumulation of BF [7,8]—an important proportion of this population is also affected by a reduced muscle mass and strength, also identified as sarcopenic obesity (SO), and this has been revealed to be a prevalent condition in all age groups in this population [9,10]. However, little is known about the alteration in BC that occurs across the lifespan of people with overweight and obesity since data on this topic are still very limited and results seem inconclusive [11].

In this context, the World Health Organization (WHO) relies on BMI to classify individuals’ adiposity based on universal cut-off points for all adults regardless their age and gender [12]. Specifically, the WHO BMI classification system considers a unique cut-off point of 30 kg/m^2^ as indicative for obesity in White, Hispanic, and Black populations for all age groups (i.e., young, middle and older adults) and for both genders (i.e., males and females) [12]. However, this traditional classification system has always been subject to criticism [13] due to several limitations [14,15], such as not fitting for all ethnicities (e.g., Asians) [16], and its inability to discriminate between the body compartments (e.g., bone, fat and muscle) [17]. In particular, recently it has been demonstrated that the fixed cut-off point to identify obesity cannot apply for all age groups, especially in middle and older age, as it is posited that a unique BMI cut-off point (i.e., 30 kg/m^2^) cannot detect changes in body compartments if this happens without significant alterations in BMI [18]; however, it is still unclear what these changes are and if there are also differences between genders.

For this reason and based on these considerations, the current study aims to compare different BC compartments (i.e., total and segmental) between three age groups and both genders, namely young, middle and older male and female adults, in a population composed exclusively of individuals affected by overweight and obesity in a clinical nutritional setting, adopting a design that matches all the age groups for weight status (i.e., body weight and BMI). We expect significant differences in BC between age groups and genders. If our hypothesis is confirmed, it will represent proof that the use of BMI and universal cut-offs used interchangeably across the different age and gender groups can be misleading in a clinical setting, and therefore alternative tools will be required.

## 2. Materials and Methods

### 2.1. Participants and Design of the Study

Our investigation is a cross-sectional observational study, which comprised individuals who are pooled from a large cohort that were initially referred to the Division of Clinical Nutrition at the Department of Biomedicine and Prevention, University of Rome “Tor Vergata” in Italy during the period from June 2018 to May 2022. The eligibility criteria for this study were having an age between 20 and 80 years, with a BMI ≥ 25 kg/m^2^, and a complete BC measurement computed by means of Dual-energy X-ray Absorptiometry (DXA). Participants were excluded if they were aged less than 20 or more than 80 years, underweight (i.e., BMI < 18.5 kg/m^2^) or normal weight according to the WHO classification (i.e., BMI < 25 kg/m^2^), or with one or more of the following conditions: pregnant or lactating, taking medication that affects body weight or composition (e.g., atypical antipsychotics, anti-epileptic drugs, glucocorticoids, β-blockers, etc.) or/and presented with medical comorbidities associated with weight loss (e.g., cancers), or severe psychiatric disorders (e.g., not well-controlled major depression).

After the satisfaction of the eligibility criteria, a matching by body weight and BMI was performed between the three main age groups to fix the effect of weight status. For the purpose of matching, the case control matching option was used at two stages, matching two age groups together each time. Matching was restricted to within a weight of 1 kg and a BMI of 0.5 units (i.e., each participant had a match in the other age group within these restrictions). At both stages, the older adults were matched to another age group. The three groups of selected matched cases were merged into one file for analysis. This resulted in a total of six separate groups consisting of three age groups (young, middle and older adults), both males and females being perfectly matched by body weight and BMI. The final sample comprised a total of 2844 individuals of both genders.

The study has been approved by the Ethics Committee of the Calabria Region Center Area Section, and a unique number has been assigned (Register Protocol No. 146 17/05/2018). All patients’ personal data were treated according to European/Italian privacy laws, and informed written consent was obtained.

### 2.2. Body Weight and Height

Body weight and height were determined by means of an electronic weighing scale (SECA 2730-ASTRA, Hamburg, Germany) and a stadiometer. The measurements were taken while the patient was in light clothes and with no shoes. The BMI of each participant was obtained using the standard formula that divides the body weight in kilograms, over the square of height in meters.

### 2.3. Body Composition and Distribution

A total body and regional (i.e., arms, trunk, and legs) BC assessment was conducted by means of a DXA (Primus, X-ray densitometer; software version 1.2.2, Osteosys Co., Ltd., Guro-gu, Seoul, Republic of Korea), which receives regular quality control and calibration before every testing session [19]. Participants were requested to not perform any type of exercise on the day that precedes the measurement. Before performing the scan, each participant was instructed to remove all clothing except for underwear, and to remove shoes, socks, and metal items before lying in a supine position on the DXA table. The whole body was scanned from the head down to the feet, with the time of measurement between 15 and 20 min, and the radiation dose of the entire procedure was estimated at nearly 0.01 millisieverts. In this study, we took into consideration the variables mentioned below:Body Fat (BF) = total body fat expressed in kg;BF percentage (BF%) (BF as a percentage of the total mass) = (BF ÷ body weight) × 100;trunk fat Trunk fat = total expressed in kg;Trunk fat percentage (%) = (trunk fat ÷ BF) × 100;Lean Mass (LM) = total lean mass, expressed in kg;Appendicular Lean Mass (ALM) = total lean mass in arms and legs, expressed in kg.

### 2.4. Statistical Analysis

A matching between the young, middle, and older groups by body weight and BMI was performed using SPSS 25 version [20]. Descriptive statistics are presented as mean and standard deviations. For mean comparison, ANOVA was used with the Bonferroni correction [21] for multiple comparisons to compare means when the equality of variance assumption was fulfilled [22], and the Welch test with the Games–Howell test were used for multiple comparisons when the equality of variance assumption was violated [23,24]. Gender-stratified multiple linear regression models were used for the association between age group and trunk fat % or ALM, while adjusting for BMI. Significance for all tests was considered at *p* < 0.05. As for multiple comparisons using Bonferroni correction, since SPSS by default calculates a corrected *p* value by multiplying the actual *p* value by the number of comparisons and compares the product to 0.05, all *p* values for Bonferroni multiple comparisons were reported as <0.05 for significance.

## 3. Results

A total of 2844 Italian adults with a BMI ≥ 25 kg/m^2^ were included in the current study, of whom 1649 (57.9%) were females and 1195 (42.0%) were males. The mean age among females was 52.5 ± 14.6 years, with 26.9% being young (31.7 ± 5.9 years), 36.5% being middle-aged (54.1 ± 4.3 years) and 36.5% being older-aged (66.3 ± 5.0 years). Across the three age categories, female participants did not differ by the mean BMI (31.1 ± 5.2 kg/m^2^ vs. 31.6 ± 4.9 kg/m^2^ vs. 31.6 ± 4.9 kg/m^2^) or weight (78.4 ± 13.8 kg vs. 78.1 ± 12.9 kg vs. 78.0 ± 12.9 kg) (Table 1).

The mean age among males was 51.8 ± 14.8 years, with 28.2% being young (31.6 ± 6.1 years), 35.9% being middle-aged (53.2 ± 4.7 years) and 35.9% being older-aged (66.2 ± 4.9 years). Across the three age categories male participants did not differ by BMI (30.3 ± 4.3 vs. 30.6 ± 4.1 vs. 30.6 ± 4.1 kg/m^2^) or weight (90.5 ± 13.9 vs. 90.2 ± 13.4 vs. 90.2 ± 13.4 kg) (Table 2).

The BC for females is presented in Table 1. The mean total BF was 35.6 ± 9.5 kg and did not differ between age categories (35.3 ± 10.4 kg vs. 35.6 ± 9.0 kg vs. 35.9 ± 9.3 kg) (Figure 1a), neither did total BF% (44.9 ± 6.4% vs. 45.5 ± 5.5% vs. 45.7 ± 5.5%) (Figure 1b). The trunk fat was significantly higher in the middle and older groups compared to young adult females (17.8 ± 5.9 kg vs. 19.1 ± 5.3 kg vs. 19.4 ± 5.4 kg; *p* < 0.05) (Figure 2a). The trunk fat percentage also followed the same trend (46.6 ± 7.4% vs. 48.2 ± 5.9% vs. 48.5 ± 5.7%; *p* < 0.05) (Figure 2b). Conversely, while total LM remained the same in the different age categories among females (39.8 ± 5.9 kg vs. 39.5 ± 5.4 kg vs. 39.8 ± 5.7 kg) (Figure 3a), the ALM was significantly lower in the middle and older groups (17.0 ± 2.6 kg and 17.1 ± 2.7 kg) compared to the young adult females (17.8 ± 2.8 kg) (*p* < 0.05) (Figure 3b).

The BC for males is presented in Table 2. The mean total BF in the overall male group was 30.5 ± 9.7 kg. Moreover, the total BF was different between age categories, with values that were significantly higher from young (29.4 ± 11.0 kg) to older adult males (31.7 ± 8.8 kg) (Figure 1a) (*p* < 0.05). Similarly for the total BF%, the lowest was observed in the young males (31.7 ± 8.3%), followed by 33.2 ± 6.1% in the middle group, and 34.6 ± 5.6% in older male adults (Figure 1b) (*p* < 0.05). In addition, the trunk fat significantly differed between age groups, with the least being in the young group (17.0 ± 6.7 kg), followed by the middle age group (18.9 ± 5.9 kg), and the highest in older male adults (19.9 ± 5.5 kg) (Figure 2a) (*p* < 0.05). The values for trunk fat percentage followed the same variation between age groups (36.6 ± 9.6% vs. 39.6 ± 6.7% vs. 41.1 ± 6.3%; *p* < 0.05) (Figure 2b). The total LM was significantly lower among the older adult males (56.5 ± 6.5 and 55.6 ± 6.6 kg) compared to the young adult males (58.3 ± 6.9 kg) (Figure 3a) (*p* < 0.05), and the ALM was significantly lower in progressive way, from 27.2 ± 3.7 kg in the young, to 25.6 ± 3.4 kg in the middle, and 24.7 ± 3.4 kg in the older male adults (Figure 3b) (*p* < 0.05).

Finally, the linear regression analysis after adjustment for BMI in line with the above findings reveals that those in the middle-aged group are more likely to have a higher trunk fat percentage by 2.66 units (1.78; 3.54) in males, and 1.23 units (0.59; 1.86) in females with respect to those in the young group. Whereas those in the older-aged group are more likely to have a higher trunk fat percentage by 4.21 units (3.34; 5.09) in males, and by 1.55 units (0.92; 2.18) in females with respect to those in the young adults (Table 3).

On the other hand, considering ALM, those in the middle-aged group are more likely to have a lower ALM by 1.7 kg (−2.16; −1.23) in males, and by 0.89 kg (−1.18; −0.61) kg in females with respect to those in the young group. Those in the older-aged group are more likely to have a lower ALM by 2.63 kg (−3.10; −2.17) in males, and by 0.81 kg (−1.09; −0.52) in females with respect to those in the young group (Table 4). Figure 4 presents an illustration of the observed difference in the trunk fat and ALM compartments by age group.

## 4. Discussion

The current study aimed to provide benchmark data on BC pattern variability between three different age groups, namely, young-, middle-, and older-aged adults of both genders, with overweight or obesity within a nutritional clinical setting. Two main findings are revealed.

### 4.1. Findings and Concordance with Previous Studies

Our main finding derives from the fact that under stable weight status (i.e., body weight and BMI), significant differences in BC patterns were revealed among the three main age groups. For instance, an individual of a determined BMI will have a different BC pattern based on the age group to whom he or she belongs. What emerges clearly from this is that relying only on BMI becomes misleading, especially in the way it is currently used with universal cut-off points applied to all individuals for all age groups and for both genders. In other words, significant differences in BC can be masked by an apparently similar BMI. In particular, the males in our study showed a higher total BF and trunk fat percentage, and a lower LM and ALM when comparing in sequence from the younger to the middle to the older age group. On the contrary, the females did not show significant differences between age groups in terms of the total BC compartments, namely, the total BF% and the total LM; however, interestingly, they showed a higher trunk fat percentage and a lower ALM from the young- to middle-aged groups that remained at similar values in the older adults with respect to the middle-aged group but in any case significantly lower than that in the young group.

Our findings are partially in accordance with one previous investigation of a similar design and population (i.e., obesity) in relation to total BC compartments for females with obesity [11]. In detail, the study by Maïmoun et al. was not initially designed for matching the age groups according to weight status; however, when the authors split their total female sample (*n* = 549) into different age groups, they obtained (by chance) no significant differences in the mean BMI, and according to the comparison, no differences were found either in BF% or in total LM [11]. On the other hand, our findings in males contrasted with those reported by the same study [11], since when the authors of that study had split their total male sample (*n* = 206), there were important differences in the mean BMI between the age groups, therefore, it was difficult to compare them based on BC compartments. Moreover, looking at the sample size of these age groups, we noticed that they were composed of small numbers of individuals that in some cases did not exceed 25–30 patients, and this makes its statistical power highly arguable [11].

In terms of difference in BC patterns between age groups (i.e., higher trunk fat and lower ALM), our results are supported—even indirectly—by a recent longitudinal case report study that included only six females with obesity who underwent a serial DXA assessment over one year and reported that BC redistribution occurred even in the case of patients who maintained their body weight (*n* = 2) [25].

It is also worth noting from our findings is that the difference in BC pattern in males seems to be more pronounced than that in females, because in the males the difference is in both total and regional compartments of fat and lean mass, while in the females their total compartments were similar (i.e., total fat and lean mass) and the differences between the age groups were reported only in terms of the regional patterns. Moreover, males revealed higher values for BF% and trunk fat percentage, and lower values for LM and ALM with older age groups sequentially from young, to middle- (higher), to the older-aged (highest) group. By contrast, in females, while the higher trunk fat percentage and lower ALM was noticed between the young to the middle-aged group, it remained similar between the middle- to the older-aged group. Therefore, our results underscore this difference between males and females; however, we are not in the position to be able to determine the reason behind this disparity between genders and how it impacts upon the changes in BC across age groups. We speculate that several factors (i.e., physical inactivity, diet, and hormonal and biomarker alterations) may have played a role in this observation. For example, an individual’s hormonal profile impacts on BC changes. For instance, testosterone is a key hormone in obesity, and low levels are associated with increased total BF and reduced total LM, which has been reported previously only in males [26,27], while interestingly it appears to have a neutral effect on these variables (i.e., BF and LM) in females with obesity [28]. Adding to this, it is well known that an age-related decline in testosterone usually occurs in males after the age of 30 at about 1% per year [29]. All in all, this appears to support our findings, and may explain indirectly why males are more vulnerable than females in terms of BC differences for total compartments.

Despite this gender disparity, both males and females displayed a similar trend in BC patterning across age groups, which was oriented toward a higher central adiposity and a lower appendicular muscle mass [30]. This can be explained through the fact that with aging, there is an increase in inflammation, which is amplified by the excess adipose tissue, that characterizes both overweight and obesity status [30]. This may lead to a redistribution of fat to the intra-abdominal area (visceral fat) and fatty infiltration into muscle [31]. Adding to all this, the lifetime weight cycling that patients with overweight and obesity are more likely to experience has been demonstrated to increase visceral fat and reduce muscle mass and strength, thereby possibly further enhancing the process of BC redistribution [32,33].

### 4.2. Potential Clinical Implications

Our findings have a number of implications. First, awareness should be raised among all healthcare professionals dealing with overweight or obesity in clinical settings to consider these differences in BC between age groups of similar BMIs, as relying on the fixed BMI cut-off without considering the content of the body compartments becomes misleading. In this direction, actions should be taken to find alternative measures as has been computed recently with a new framework by the European Association for the Study of Obesity (EASO) which takes into account other indicators in addition to BMI [34]. Second, our findings highlight the variability in BC compartments between males and females with overweight or obesity. Gender-specific clinical outcomes are a hot research topic at present and are garnering increased academic interest [35,36].

### 4.3. Study Strengths and Limitations

Our findings have several strengths. Firstly, it is one of the very few studies to investigate the differences in BC and its distribution between gender and age groups in a large sample of patients with overweight or obesity in a real-world, outpatient nutritional setting. Secondly, BC was measured using DXA, a tool that has been validated since it guarantees a reasonable precision and reliability for the assessment of the two main body components, namely lean and fat mass in individuals with obesity [37,38]. Indeed, its use for this aim has been clearly recommended by the International Society of Clinical Densitometry [39]. In addition, DXA with respect to high precision techniques such as the computed tomography (CT) scan and magnetic resonance imaging (MRI), remains the most widely available method in real-world clinical practice including nutritional settings. CT scans and MRI, however, are expensive and not usually available in such settings [40]. Moreover, DXA for BC assessment is relatively cheap, easy-to-use, fast and safe for both regional and total BC compartments. It does not require highly trained technicians or extended testing time and delivers a very low dose of radiation that is usually attenuated. By contrast, CT scans and MRI usually provide only segmental BC measurements (i.e., visceral adipose tissue [VAT], subcutaneous adipose tissue [SAT], etc.), and obtaining the total BC compartments (i.e., total LM and FM) is not practical. Additionally, the high radiation dose of CT scans means its use for purely BC assessment is limited and reserved only for patients who are undergoing CT for another clinical indication [40]. Thirdly, the study design matched the three main age groups for BMI and body weight, and this is considered a strength since it enables detection of any potential differences in BC compartments in the absence of a significant difference in body weight status. However, our study also has certain limitations. Firstly, data were collected in a single unit and our results thus require external validation across other populations [41] since there are race/ethnic differences in the relationship between BMI and BC compartments [42]. Secondly, our study is cross-sectional, therefore at best it can only reveal the differences in BC variables between age groups, and no information regarding the changes that occur in the same age group over time were included for which a longitudinal assessment is required [43]. Finally, we had no information regarding lifestyle habits, smoking, physical activity levels, dietary intake, or biochemical and hormonal blood tests, or current or previous health status, which are factors known to affect or relate to BC [44].

### 4.4. Future Research

Firstly, other investigations should replicate our findings to confirm the results in other European countries and in populations of other ethnicities. Secondly, research needs to clarify the factors that are involved in the differences in BC in both males and females affected by overweight and obesity so as to explain the gender discrepancies. Thirdly, studies are required to assess the impact of these differences in BC patterns on health outcomes (medical and psychosocial), as well as on longevity and mortality. Finally, new tools should be developed to overcome the limits of BMI. These need to be able to screen the increase in BF, especially in the central body regions, as well as the reductions in LM which are foremost in the extremities as an expression of muscle mass.

## 5. Conclusions

Our study highlighted that individuals with overweight or obesity display a significant difference in BC patterns, namely a higher truncal accumulation and a lower ALM. The central obesity and lower appendicular muscle mass may have negative health consequences (i.e., insulin resistance, inflammation, high fall risk) in the absence of any meaningful differences in BMI. Accordingly, the use of the latter becomes obsolete and misleading, and future research is needed to identify new tools that are able to detect these changes in this specific population.

## Figures and Tables

**Figure 1 nutrients-16-02415-f001:**
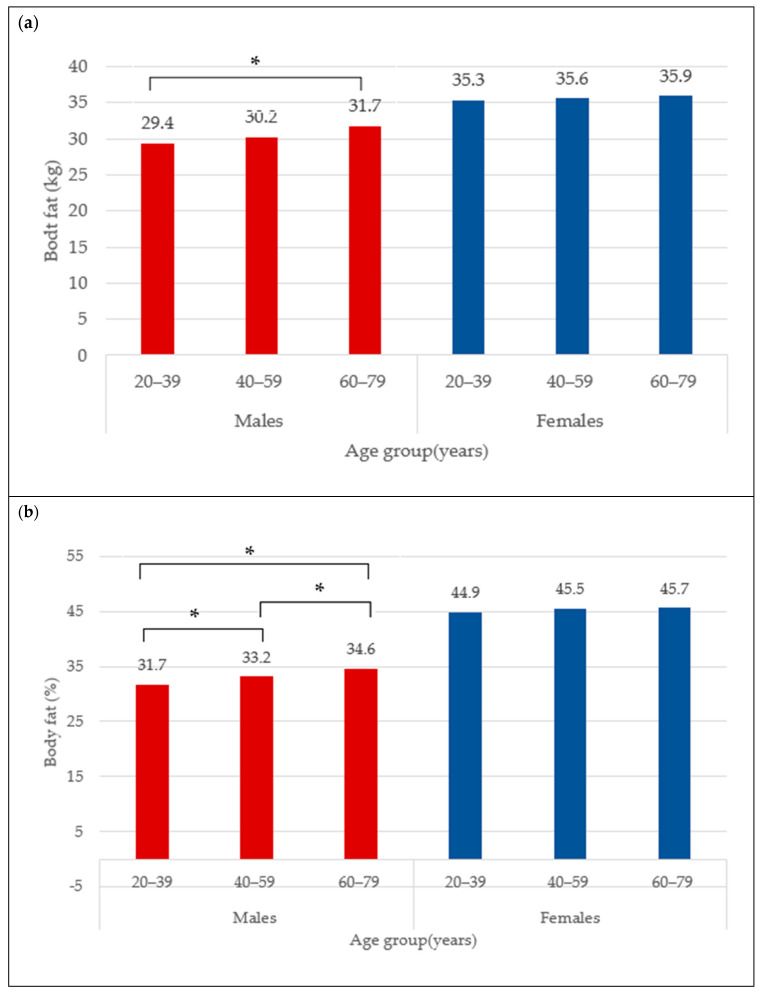
Body fat (kg) (**a**) and body fat percentage (**b**) across age groups in males and females. * indicates mean difference at *p* < 0.05.

**Figure 2 nutrients-16-02415-f002:**
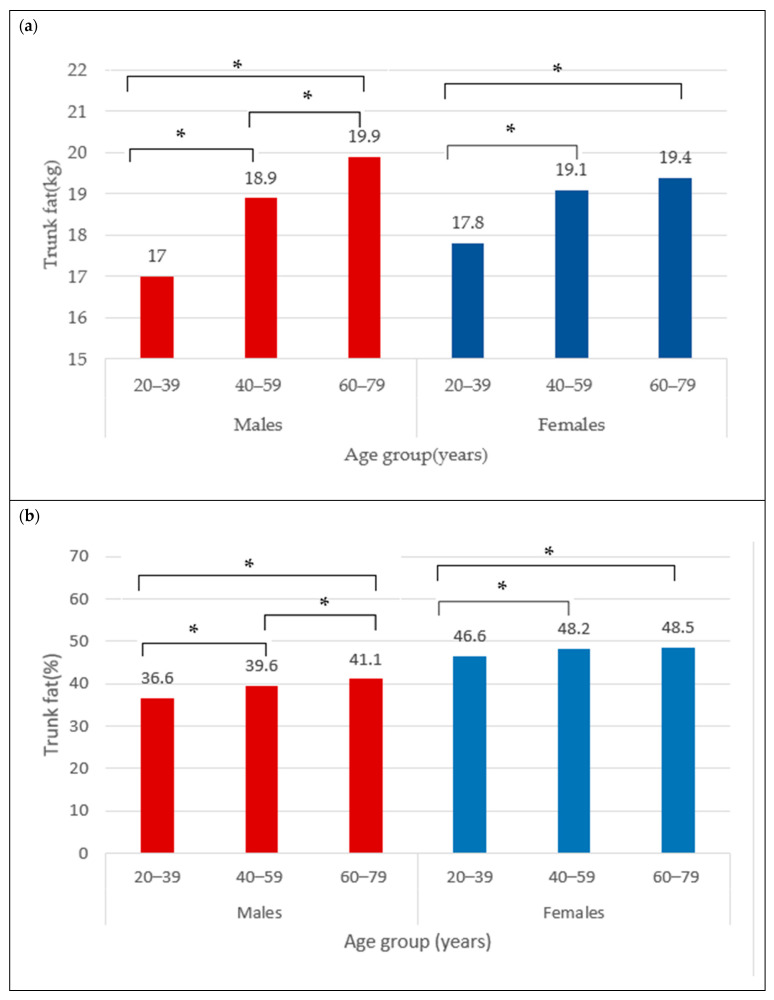
Trunk fat (kg) (**a**) and trunk fat percentage (**b**) across age groups in males and females. * indicates mean difference at *p* < 0.05.

**Figure 3 nutrients-16-02415-f003:**
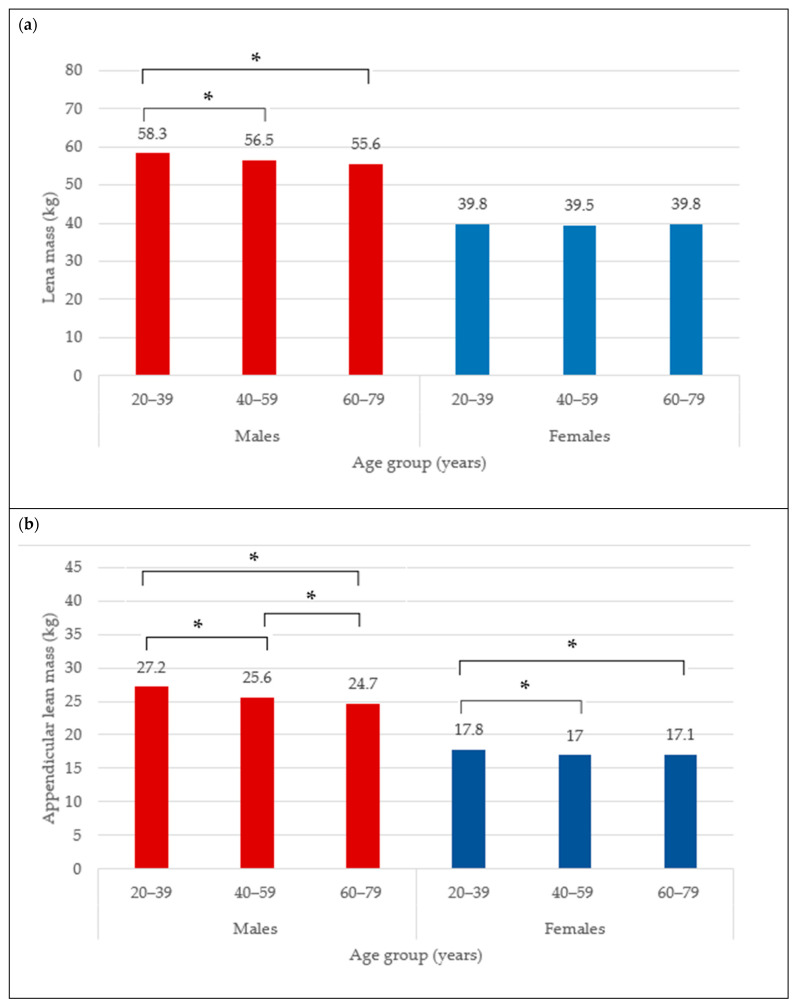
Lean mass (kg) (**a**) and appendicular lean mass (**b**) across age groups in males and females. * indicates mean difference at *p* < 0.05.

**Figure 4 nutrients-16-02415-f004:**
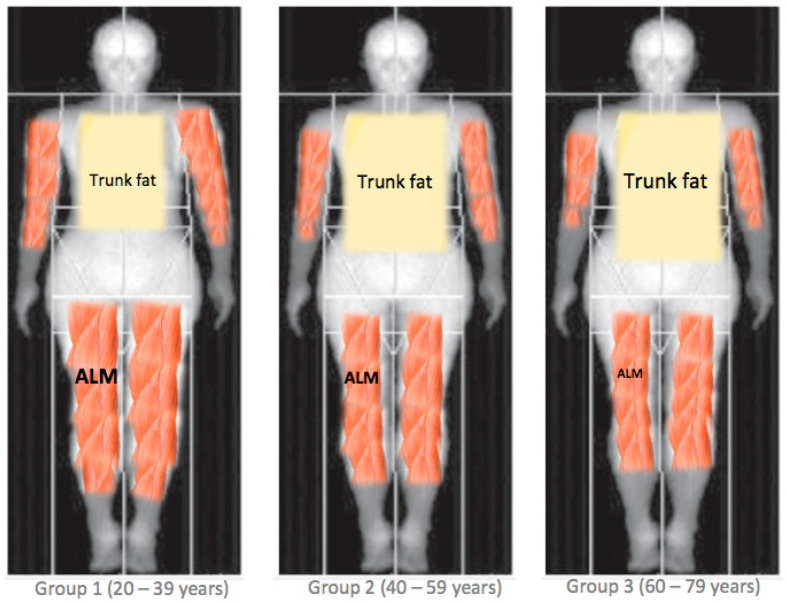
The proposed differences in body composition patterns between age groups. ALM = appendicular lean mass.

**Table 1 nutrients-16-02415-t001:** Age, anthropometric and BC characteristics of females across age groups (*n* = 1649).

		Age Group			
	Total (*n* = 1649)	20–39 Years (*n* = 445)	40–59 Years (*n* = 602)	60–79 Years (*n* = 602)	Significance
Age years	52.5 (14.6)	31.7 (5.9) ^a^	54.1 (4.3) ^b^	66.3 (5.0) ^c^	*p* < 0.05 ^§^
Weight (kg)	78.1 (13.2)	78.4 (13.8)	78.1 (12.9)	78.0 (12.9)	*p* > 0.05 ^¥^
BMI (kg/m^2^)	31.4 (5.0)	31.1 (5.2)	31.6 (4.9)	31.6 (4.9)	*p* > 0.05 ^¥^
BF (kg)	35.6 (9.5)	35.3 (10.4)	35.6 (9.0)	35.9 (9.3)	*p* > 0.05 ^¥^
BF% (%)	45.4 (5.8)	44.9 (6.4)	45.5 (5.5)	45.7 (5.5)	*p* > 0.05 ^§^
Trunk fat (kg)	18.9 (5.5)	17.8 (5.9) ^a^	19.1 (5.3) ^b^	19.4 (5.4) ^b^	*p* < 0.05 ^¥^
Trunk fat (%)	47.9 (6.3)	46.6 (7.4) ^a^	48.2 (5.9) ^b^	48.5 (5.7) ^b^	*p* < 0.05 ^§^
LM (kg)	39.7 (5.7)	39.8 (5.9)	39.5 (5.4)	39.8 (5.7)	*p* > 0.05 ^§^
ALM (kg)	17.3 (2.7)	17.8 (2.8) ^a^	17.0 (2.6) ^b^	17.1 (2.7) ^b^	*p* < 0.05 ^¥^

Values are means and SD. BC = body composition; BMI = body mass index; BF: body fat; BF% = body fat percentage; LM = lean mass; ALM = appendicular lean mass. Multiple comparison *p* values for Welch’s test ^§^ or ANOVA ^¥^; ^a,b,c^ values with different superscripts are significantly different at *p* < 0.05.

**Table 2 nutrients-16-02415-t002:** Age, anthropometric and BC characteristics of males across age groups (*n* = 1195).

		Age Group			
	Total (*n* = 1195)	20–39 Years (*n* = 337)	40–59 Years (*n* = 429)	60–79 Years (*n* = 429)	Significance
Age years	51.8 (14.8)	31.6 (6.1) ^a^	53.2 (4.7) ^b^	66.2 (4.9) ^c^	*p* < 0.05 ^§^
Weight (kg)	90.3 (13.6)	90.5 (13.9)	90.2 (13.4)	90.2 (13.4)	*p* > 0.05 ^¥^
BMI (kg/m^2^)	30.5 (4.1)	30.3 (4.3)	30.6 (4.1)	30.6 (4.1)	*p* > 0.05 ^¥^
BF (kg)	30.5 (9.7)	29.4 (11.0) ^a^	30.2 (9.2) ^a,b^	31.7 (8.8) ^b^	*p* < 0.05 ^§^
BF% (%)	33.3 (6.8)	31.7 (8.3) ^a^	33.2 (6.1) ^b^	34.6 (5.6) ^c^	*p* < 0.05 ^§^
Trunk fat (kg)	18.7 (6.1)	17.0 (6.7) ^a^	18.9 (5.9) ^b^	19.9 (5.5) ^c^	*p* < 0.05 ^§^
Trunk fat (%)	39.3 (7.7)	36.6 (9.6) ^a^	39.6 (6.7) ^b^	41.1 (6.3) ^c^	*p* < 0.05 ^§^
LM (kg)	56.7 (6.8)	58.3 (6.9) ^a^	56.5 (6.5) ^b^	55.6 (6.6) ^b^	*p* < 0.05 ^¥^
ALM (kg)	25.7 (3.6)	27.2 (3.7) ^a^	25.6 (3.4) ^b^	24.7 (3.4) ^c^	*p* < 0.05 ^¥^

Values are means and SD. BC = body composition; BMI = body mass index; BF: body fat; BF% = body fat percentage; LM = lean mass; ALM = appendicular lean mass. Multiple comparison *p* values for Welch’s test ^§^ or ANOVA ^¥^; ^a,b,c^ values with different superscripts are significantly different at *p* < 0.05.

**Table 3 nutrients-16-02415-t003:** Linear regression coefficients for the association between trunk fat % and age group while adjusting by BMI among males and females.

	Males	Females
	β (95%CI)	
BMI (kg/m^2^)	1.04 (0.96; 1.13)	0.73 (0.68; 0.78)
Middle-aged adult group	2.66 (1.78; 3.54)	1.23 (0.59; 1.86)
Older-aged adult group	4.21 (3.34; 5.09)	1.55 (0.92; 2.18)

BMI = body mass index.

**Table 4 nutrients-16-02415-t004:** Linear regression coefficients for the association between ALM and age group while adjusting by BMI among males and females.

	Males	Females
	β (95%CI)	
BMI (kg/m^2^)	0.31 (0.27; 0.36)	0.27 (0.25; 0.30)
Middle-aged adult group	−1.70 (−2.16; −1.23)	−0.89 (−1.18; −0.61)
Older-aged adult group	−2.63 (−3.10; −2.17)	−0.81 (−1.09; −0.52)

BMI = body mass index.

## Data Availability

The dataset in the present study is available upon request.

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
