# Peer review of "Difference in Body Composition Patterns between Age Groups in Italian Individuals with Overweight and Obesity: When BMI Becomes a Misleading Tool in Nutritional Settings"

_nutrients, 2024, doi:10.3390/nu16152415_

Round 1

Reviewer 1 Report

Comments and Suggestions for Authors

1. While the authors claim that they are using the Bonferroni correction, it’s not easy to understand how, since they don’t report p-values, only if the value is above and below 0.05. In the statistical results section you could include the exact p-value (e.g., “The mean difference was significant (p = 0.023)”). If you have applied adjustments for multiple comparisons, please also clarify the adjusted significance threshold and indicate whether the p-value falls below this threshold (e.g., “After Bonferroni correction, p = 0.023, below the adjusted threshold of p < 0.005”). 

2    How is reference 24 relevant?

3     Please discuss DXA-scans in comparison to CT or MRI for body composition.

4     This is a cross-sectional study, and despite this the authors routinely talk about changes over time. For instance, this passage “Our study highlighted that across the lifespan of an individual with overweight or obesity there is a redistribution of BC compartments, namely a truncal accumulation and a decrease in ALM.”, or “the changes across lifespan appear to be different between males and females in a population of individuals with overweight or obesity. This is particularly misleading due the BMI matching between age-groups. The old-age groups are unlikely to have had a the same BMI when they were young, and the current young-age groups are likely to have a higher BMI when they reach old age.

Please emphasize the cross-sectional nature and use terminology like "pattern," or "association," which are preferable in cross-sectional studies to describe findings without implying causality or changes over time.  Also, discuss this more thoroughly as a weakness, especially since no data on potential confounders (changes in diet, smoking, physical activity likely differs in Italy over the past 50 years).

5.      Please make a flow chart for exclusions clearly showing the initial number of participants, and step by step exclusions as percentages and the number censored for each individual exclusion criteria, and the matching process. Also please define which drugs are an exclusion criterion, not only ‘medication that affects body weight or composition’.

The statement on BMI being a “hot topic” includes references to research that is 20 years old (references 26-27). Please cite more recent research, e.g. https://www.nature.com/articles/s41591-024-03095-3.

6.     The discussion is poorly written and sometimes confusing. There are complex winding sentences where it’s unclear what the authors are referring to and numerous parentheses are used to clarify the poor writing instead of writing clearly to begin with. For example, the paragraph about reference 11 reads as if the authors are discussing their own study “This study” instead of “in that study”.

7.    Don’t unnecessarily cite yourself (references nr 9, 18 and 34).

8. The definition of sarcopenia is not muscle loss in the extremities, it is general low muscle mass.

Comments on the Quality of English Language

Need to improve.

Reviewer 2 Report

Comments and Suggestions for Authors

A key message of this paper is that at the same level of BMI, body composition differs by age and sex.  This is well known, and one of the key reasons for critiques of BMI as an indicator of overweight or obesity.  However, this paper is useful in providing specific evidence for the magnitude of differences that characterize age groups. With appropriate revision, this research may be of relevance and interest to a wide audience including researchers and clinicians, specifically those focused on OW/OB or bariatric patients. The findings from this work will hopefully encourage replications and provide the research community interested in body composition and tailored anthropometric cut-points with further insight on discrepancies of body composition throughout life. 

A key problem is the presentation and discussion of results as if they reflect changes in body composition with age rather than differences by age which are demonstrated from age-stratified cross sectional data.  In the absence of longitudinal data, we can’t tell whether the differences are age-related, or the result of cohort effects or secular trends.  The data in figures 1-3 should not be presented as line graphs which suggest longitudinal data, and language should be appropriate for the cross sectional data.

Line 125 and following: It is hard to understand what is meant by the “matching between the young, middle, and older groups by body weight and BMI 125 was performed using SPSS 25 …”  Exactly how was this done? This is a key aspect of methods that readers must understand to interpret the results.

All of the numbers/results presented in the text are redundant with what is presented in the table and make it more tedious to read.

The current writing mechanics and conventions may distract readers. Sentences are way too long, verb-tense agreement issues, misuse of some phrases or words, and repetitive wording need to be addressed to meet the quality of the journal. 

Findings seem overstated given the cross-sectional study design with no additional information pertaining to lifestyle or current health status.  Methods 

The race/ethnic demographics of the sample should be noted, considering previous literature on ethnic differences with BMI.  

DXA: while DXA is recognized as the gold standard for body composition analysis, more information should be provided about about the precision and reliability of DXA measurements in your study population.

Other minor wording issues: be consistent in the use of sex vs gender and in use of present vs past tense  

Comments on the Quality of English Language

needs editing for language, consistency

Round 2

Reviewer 2 Report

Comments and Suggestions for Authors

no additional comments